# Antimicrobial Peptide Mechanisms Studied by Whole-Cell Deuterium NMR

**DOI:** 10.3390/ijms23052740

**Published:** 2022-03-01

**Authors:** Sarika Kumari, Valerie Booth

**Affiliations:** 1Department of Biochemistry, Memorial University of Newfoundland, St. John’s, NL A1C 5S7, Canada; skumari@mun.ca; 2Department of Physics and Physical Oceanography, Memorial University of Newfoundland, St. John’s, NL A1C 5S7, Canada

**Keywords:** whole cell NMR, antimicrobial peptides (AMPs), host defense peptides, HDPs, MSI-78, pexiganan

## Abstract

Much of the work probing antimicrobial peptide (AMP) mechanisms has focussed on how these molecules permeabilize lipid bilayers. However, AMPs must also traverse a variety of non-lipid cell envelope components before they reach the lipid bilayer. Additionally, there is a growing list of AMPs with non-lipid targets inside the cell. It is thus useful to extend the biophysical methods that have been traditionally applied to study AMP mechanisms in liposomes to the full bacteria, where the lipids are present along with the full complexity of the rest of the bacterium. This review focusses on what can be learned about AMP mechanisms from solid-state NMR of AMP-treated intact bacteria. It also touches on flow cytometry as a complementary method for measuring permeabilization of bacterial lipid membranes in whole bacteria.

## 1. Introduction to Antimicrobial Peptides

Antimicrobial peptides (AMPs) are a diverse class of molecules. There have been thousands of AMP sequences reported to date [1]. Most AMPs share characteristics of short amino acid chain length, positive overall charge, structures with both hydrophobic and hydrophilic regions, and selectivity for pathogens over host cells [2]. Given the number and variety of them, it is helpful to organize AMPs into categories. They can be classified in a variety of ways, including by structure, amino acid sequence, and biological function. One basic division is between linear AMPs and AMPs stabilized by intra-chain disulfide bridges [3]. The first group includes linear peptides with an α-helical structure, such as magainin and cecropin; as well as peptides rich in particular amino acids, such as glycine, proline, arginine, tryptophan, and histidine. Most of these peptides are unstructured in aqueous solution but become structured when in contact with lipids, trifluoroethanol, and detergent micelles [4]. The second group consists of cysteine-containing polypeptides that form disulfide bridge(s), such as insect defensins [5]. When discussing AMP structure–function relationships, it is also useful to divide them up via secondary structure—i.e., α-helical, beta-sheet, and extended coil—each of which is discussed in turn below.

The α-helical AMPs were the first AMP structure class to be characterized [6], and have since been extensively studied. One good example of α-helical AMPs are the magainins isolated from the African clawed frog *Xenopus laevis*, which are active against Gram (+) and Gram (−) bacteria, fungi, yeast, and viruses [7]. The structure–function relationships of magainins have been probed extensively [8]. As for most helical AMPs, amidation of the C-terminus of magainins enhances the electrostatic interaction between the positively charged peptide and the negatively charged bacterial membrane. This interaction stabilizes the α-helical structure at the membrane interface [9]. Magainin was the first AMP to be tested in the clinic but failed in clinical trials because it was not better than standard treatment [10]. However, the C-terminally modified MSI-78 (pexiganan) peptide, an analog of magainin 2 with more positive charge, is currently in clinical trials as a topical antimicrobial treatment for mild-to-moderate diabetic foot ulcers [11,12].

The second secondary structure group of AMPs has β-strands. These peptides adopt a β-sheet structure when in contact with a lipid membrane. In contrast with α-helical AMPs, the structure of these β-sheet peptides is less flexible because of the structural restraints introduced by the disulfide bonds between the β-strands. This is the case, for example, with tachyplesin, protegrin, and human α-defensins [10,13]. Defensins are a large group of AMPs involved in antibacterial, antifungal, antiviral, immune, and inflammatory responses [14].

The third secondary structure group of AMPs are those that form neither α-helices nor β-sheets. For example, the cathelicidin family [15] is rich in proline, an amino acid known to break α-helical and β-sheet secondary structure. The cathelicidin LL-37 is named for its 37 amino acids and N-terminal di-Leucine (Leu) (LL) motif. LL-37 is active against Gram (−) and Gram (+) bacteria, including *Escherichia coli* (*E. coli*), *Staphylococcus aureus* (*S. aureus*), and *Pseudomonas aeruginosa* [16]. Another example is indolicidin from bovine neutrophils, which is rich in tryptophan and has only 13 amino acids [17]. Nuclear magnetic resonance (NMR) and circular dichroism (CD) studies reveal that indolicidin forms a well-defined extended structure in the presence of membrane-mimicking micelles [17,18].

## 2. Mode of Action of AMPs

AMPs interact with membranes, either to: (1) harm the target cell directly by permeabilizing the cell membrane; or (2) cross the membrane and reach an intracellular target [2,19].

### 2.1. Direct Killing of Target Cells by Permeabilizing the Membrane

AMP-induced membrane permeabilization is a widely accepted mechanism of action of AMPs [13,20]. They bind to the lipid bilayer, driven by electrostatic and hydrophobic interactions [2,21]. Since bacterial membranes are composed of a phospholipid bilayers with a high proportion of anionic lipids compared to mammalian cells, there are relatively strong electrostatic interactions between AMPs and bacteria [22]. A variety of models have been used to describe the action of AMPs once they bind the membrane, including the barrel-stave model, the toroidal pore model, the disordered toroidal pore model, and the carpet model [23].

In the barrel-stave model AMPs are positioned in a barrel-like ring around an aqueous pore. The hydrophobic face of the AMP points outwards from the pore so that it is in contact with the lipid acyl chains of the membrane, while the hydrophilic AMP faces form the inner lining of the pore [2,24]. Only a few AMPs—for example, alamethicin [2], and possibly pardaxin [25]—are thought to form barrel-stave channels.

The toroidal pore is one of the best-characterized AMP-membrane disruption mechanisms. AMPs bind in the polar head group region of the lipids, pushing the headgroups apart and inducing a positive curvature strain [26,27,28]. This results in a pore lined by both AMPs and, unlike barrel-stave pores, lipid polar headgroups. Both toroidal and barrel-stave pores lead to membrane depolarization and, consequently, cell death. Several AMPs such as aurein 2.2 [29] and melittin [30] have been shown to form toroidal pores. An updated version of the toroidal pore is the disordered toroidal pore model. In this model, the AMP-lined pore is not well organized and has an irregular, dynamic arrangement [27,31].

In the carpet model, AMPs act without forming specific pores in the membrane [32,33]. Rather, the AMPs accumulate parallel to the lipid bilayer and reach the surface concentration needed to envelop the surface of the membrane, thereby forming a ‘carpet’. This accumulation of many peptides destabilizes the membrane [28].

### 2.2. Killing of Bacteria through Non-Membrane-Permeabilizing Mechanisms

Almost all AMPs have a high affinity towards the cytoplasmic membrane, which leads to at least a certain amount of membrane perturbation. That said, there is a growing list of AMPs that have been shown to harm bacteria without disrupting the membrane enough to cause substantial permeabilization. Such peptides generally cross the membrane and reach one or more intracellular targets [34,35,36]. These AMPs interact with the cytoplasmic membrane first and then accumulate intracellularly to block cellular processes. Some intracellular-acting AMPs can interact with DNA or RNA directly, thus interfering with their replication, translation, and synthesis processes [34,35]. Defensins can block cell wall synthesis [37]. Buforin II translocates across the lipid bilayer to bind to DNA and RNA without causing cell lysis [38,39]. MSI-78 binds and destabilizes ribosomes [40] and PR-39 has also been suggested to target intracellular by preventing DNA and protein synthesis [41].

### 2.3. AMPs vs. HDPs vs. CPPs

AMPs are increasingly being referred to as ‘host defence peptides’ (HDPs), a term that captures the more general mechanisms of some AMPs/HDPs, for example, in modulating the host’s immune response [42]. Cell-penetrating peptides (CPPs) are another class of membrane-active peptides that share similar physiochemical properties with AMPs. Like AMPs, CPPs interact with membranes, but in contrast to AMPs, CPPs do not permeabilize the membrane [43]. Instead, CPPs translocate from one side of the bilayer to the other without bilayer permeabilization [44]. Both CPPs and AMPs have attracted attention due to their potential in novel drug delivery systems [45].

Cell-penetrating peptides are found in nature, are typically quite short peptide sequences, and can be linked to cargo for transport into cells [45]. CPPs can deliver a variety of molecules into cells—including proteins, peptides, siRNA, DNA, liposomes, and nanoparticles [46]—leading to much interest in their potential clinical uses [47]. CPPs enter cells by one of two modes, either by endocytosis, which is energy-dependent, or by energy-independent passive uptake [47]. In both modes, peptides adsorb at the membrane surface, where they interact with negatively charged lipids, and some perhaps with glycoconjugates or membrane proteins [48]. One challenge in developing CPPs as delivery systems is that they often toxically permeabilize cells beyond a safe-threshold concentration [43]. Although CPPs have potential for use in drug delivery, their ability to enter cells of almost any kind still confers significant toxicity concerns that must be addressed.

### 2.4. AMP Interactions with Non-Lipid Cell Envelope Components of Bacteria

Knowing the structure of the non-lipid components of the cell envelope is important for understanding how AMPs traverse them to reach the target cell membrane. In addition to the lipid bilayer, bacteria cell envelopes can have peptidoglycan (PGN), teichoic acid (TA), and lipopolysaccharide (LPS), as well as membrane proteins. Gram (+) bacteria (Figure 1A) have a single lipid bilayer surrounded by a thick PGN layer with negatively charged TAs anchored to the PGN. By contrast, Gram (−) bacteria (Figure 1B) are surrounded by two lipid bilayers with a thin PGN layer between them. The outer leaflet of the outer layer of Gram (+) bacteria is composed mainly of LPS, with the carbohydrate moieties of the LPS facing outwards.

How AMPs initially traverse the non-lipid components of bacterial cell envelopes to reach the bilayer is still poorly understood. This is because most studies of AMP mechanism have focussed on AMPs interacting with model lipid membranes, for example using fluorescence-based permeabilization assays of vesicles or NMR of AMPs in liposomes [49,50]. One way to illustrate the potential importance of the non-lipid components is to compare the molar AMP-to-lipid (AMP:L) ratio needed to see permeabilization of synthetic liposomes with the AMP:L ratio needed to see AMP activity in actual cells. In general, more AMP is needed to see activity in cells, suggesting the AMP binds cell components beyond the lipids.

One standard way to measure AMP activity against cells is by establishing the minimal inhibitory concentration (MIC), the minimum AMP concentration needed to prevent cells from growing. Several researchers have estimated or measured the AMP:L ratio at the MIC in whole cells and compared the values to those typical from in vitro experiments in model lipid vesicles. An early estimate proposed that, in liposomes, the bound AMP:L ratio is about 1:200 [2]. In stark contrast, in bacteria, the bound AMP:L ratio is about 10–100:1. An alternate approach by Melo et al. [51] used the partition constant to understand the relationship between liposome and bacterial experiments. Their in vitro and in vivo data for two AMPs, melittin and omiganan, indicated the cell-bound AMP:L ratio was 2.3 to 9.2 times higher than the threshold to see effects in liposomes. In a more direct approach, the Stella group [52] has developed an experimental approach using a special minimal medium where the bacteria are metabolically active but do not multiply. This has allowed them to investigate bactericidal activity against *E. coli* and AMP–cell association and showed that, at the minimum bactericidal concentration (MBC), 10^7^ fluorescently labelled AMP molecules are bound to each cell, i.e., AMP:L ratio of ~1:3 to 5:1.

These studies suggest that AMPs may bind to molecules present in bacteria that are not present in liposomes. For Gram (−) bacteria, several studies indicate that AMPs interact with the LPS layer of the bacterial cell envelopes. Experiments on *E. coli* mutants where the LPS layer was absent increased the effectiveness of seven different AMPs, indicating that the LPS layer protects the bacteria from AMP [53]. Such interactions between the AMPs and the LPS in the cell envelope of Gram (−) bacteria need to be accounted for to provide a complete view of AMPs’ mechanism of action.

Turning to Gram (+) bacteria with their thick PGN layer, it has been proposed that PGN does not prove to be a barrier for many AMPs given PGN’s lack of negative charge [54]. On the other hand, the AMP eosinophilic cationic protein has been shown to have strong interactions with both LPS and PGN using a fluorescent displacement assay [55]. Considering the importance of electrostatic interactions between positively charged AMPs and their targets, the negatively charged TA component of Gram (+) bacterial cell envelopes has been proposed to attract AMPs, sequestering them away from the lipid membrane and thus protecting the cells [56].

## 3. Extending Biophysical Techniques That Probe AMP Mechanism from Model Membranes to Whole Cells

The activity of AMPs is commonly assessed by assays with cells, such as determining the MIC, i.e., the minimal concentration of AMP needed to prevent bacteria from growing [57]. By contrast, most investigations into the mechanism of AMP membrane perturbation, including NMR and fluorescence [20,58,59], employ only lipids (Figure 1C) rather than the entire bacterium (Figure 1A,B). ^2^H NMR of lipids deuterated all along their acyl chains gives valuable information on the structure and dynamics at specific locations along the acyl chain (Figure 1C) [60] and how these are affected by an AMP. Complementarily, ^31^P NMR (Figure 1C), can be used to assess the behaviour of the lipid headgroups with and without AMPs [61]. For investigating at the AMP itself, rather than its effects on the lipids, NMR of peptides labelled with ^15^N on the peptide backbone and/or ^2^H nuclei in alanine sidechain methyls provides information on AMP structure and orientation within the bilayer [8,62]. Such methods applied to AMPs in model membranes have been tremendously important in providing data to support the AMP-induced bilayer disruption models described in Section 2.1 above.

However, to better understand AMP mechanisms in the context of the non-lipid interactions described in Section 2.2 and Section 2.4, it is helpful to extend the model lipid studies of AMP mechanisms to whole cells. This provides the full complexity of molecules that AMPs may interact with and that likely modify an AMP’s membrane-perturbing actions. The following section will give examples of our and our colleagues’ work on solid-state NMR of AMPs in intact bacteria, with a particular focus on ^2^H NMR methods applied to reveal different aspects of AMP mechanisms of action. In addition to the studies highlighted below, the reader is directed to other works for a broader view of in-cell NMR, including the exciting developments in solution NMR [63,64] and solid state ^13^C NMR [65,66]. For the following discussion, it is helpful to know that solid-state NMR can be applied to samples like bacteria in two primary ways—static NMR and magic angle spinning NMR. The two approaches provide similar information on AMP-induced bilayer perturbations.

The Davis group obtained the first ^2^H-NMR spectra of membrane-deuterated bacteria in the early 1980s [67]. The first application of whole bacteria ^2^H-NMR to AMPs came in 2012 [68], where the AMP MSI-78 was shown to drastically impact intact cells’ lipid acyl chain order. This work employed a modified strain of *E. coli,* which could not metabolize or synthesize the fatty acids and thus incorporated into cell membranes high levels of deuterated acyl chains from deuterated palmitic acid (PA) provided in the growth media [68,69]. Shortly after this, the Marcotte group developed a method for membrane-deuterating bacteria without employing mutants by adding deuterated PA in complex with dodecyl phosphocholine (DPC) to the growth media [70]. In this and subsequent work, several factors have been identified that support the acquisition of reproducible ^2^H-NMR spectra from the bacteria, including adjusting the relative amounts of palmitic and oleic acids for the type of bacteria being grown, being very consistent with the growth and harvesting protocols used, transferring the cells into the NMR spectrometer quickly after growth, acquiring spectra in sequential blocks to monitor for changes in spectra over time, and using cell viability assays to assess how many bacteria are alive and metabolizing after their time in the NMR spectrometer [68,69].

With both the initial studies of AMPs in whole deuterated bacteria [68,70], it was exciting to see how the ^2^H experiments traditionally done with AMPs in model lipids could be recapitulated in the context of whole bacteria. Before examining how whole cell ^2^H NMR speaks to the membrane perturbing mechanisms of AMPs, we first discuss the information contained in ^2^H NMR spectra of membanes in general.

^2^H NMR spectra of lipids in model membranes or membrane-deuterated bacteria encode information about the lipid acyl chain dynamics at various positions along the chain. The more constrained the motion at a particular carbon–deuteron bond on the acyl chain, the wider the ^2^H NMR splitting will be for that chain position. Thus, the prominent edges at ±12.5 kHz (Figure 2A) are dominated by the acyl chain deuterons closest to the lipid head groups that have the most constrained motions. On the other hand, deuterons nearer to the methyl end of the lipids, and thus nearer the center of the bilayer, have much freer motions and thus contribute intensity nearer to the center of the NMR spectra [60]. Hence, when a lipid-membrane-perturbing AMP is added to the sample, this is commonly seen as a change in the shape of the NMR spectra with intensity transferred from the outer edges (that correspond to more constrained motion) to nearer the center of the spectra (that indicate less constrained motion).

The ^2^H NMR spectra of membrane-deuterated Gram (−) *E. coli* bacteria are substantially altered by treating the bacteria with the AMP MSI-78 (pexiganan) (Figure 2A). To facilitate comparison with both MIC assays and model lipid studies, the amount of MSI-78 in the whole-cell NMR is expressed as a weight % of the dry weight of bacteria. Interestingly, much more (~30 times more) MSI-78 per lipid needs to be added to the whole bacteria samples to see the same lipid disruption in the corresponding model lipid NMR studies [68,71]. This observation is consistent with some of the AMP binding non-lipid targets and/or the presence of the non-lipid components of the cell envelope protecting the lipid bilayer from AMP-induced disruption.

Gram (+) bacteria, *B. subtilis*, treated with AMPs CAME and BP100, have also been studied with whole-cell NMR [70,71,72,73]. This work adjusted the ratio of deuterated PA to unlabelled oleic acid in the media to keep the cells healthy. AMP-induced changes similar to those seen in the deuterium NMR spectra of *E. coli* were also observed in *B. subtilis,* suggesting similar lipid disruption mechanisms for the three peptides and similar effects on both Gram (+) and Gram (−) bacteria. Whole-cell deuterium NMR employing the AMPs caerin 1.1 and aurein 1.2 has been performed with both Gram (−) bacteria *E. coli* and Gram (+) bacteria *B. subtilis* [74]. Comparison of the results in the two types of bacteria demonstrated that at the same AMP concentration, the AMPs disrupted *B. subtilis* less than *E. coli.* Based on this comparison, the authors suggested that cell wall components present in *B. subtilis*—such as TA and PGN—interact with the AMPs, reducing their local concentration at the lipid membrane.

While visual inspection of the deuterium NMR spectra allows us to draw many of the conclusions above, it is useful to quantify the shape of the spectra in terms of spectral moments, M1, M2, and Δ_2_ [68,69,74]. Computing these values from the spectra facilitates comparisons and allows experiments performed with static solid-state NMR to be compared with results from magic angle spinning (MAS) NMR. MAS NMR has the significant advantage of much faster acquisition times, although it is not yet clear if it contains the same information as static NMR since some of the observed parameters seem to depend on spinning rate [75]. The first and second moments, M1 and M2, are the frequency and squared-frequency-weighted averages of the spectral intensity and reflect the average order parameter of the acyl chains (Figure 2A) [60,73]. In other words, the greater the moments, the more ordered the acyl chains are. On the other hand, Δ_2_ reflects the shape of the spectra and is particularly useful to indicate when a treatment perturbs one region of the acyl chain more than other regions of the acyl chain. AMP treatments of deuterated bacteria make dramatic changes in Δ_2_, which are consistent with the AMPs causing the lipid acyl chain regions closest to the lipid headgroups to become much less motionally constrained than in untreated cells. This is in general agreement with those models of AMP lipid disruption mechanism discussed in Section 2.1 that involve AMPs binding in the headgroup regions of the lipids.

^2^H NMR of intact cells provides complementary information to other techniques. For example, dye-leakage assays report indirectly on bilayer characteristics by indicating if the bilayer is perturbed enough to allow a dye to cross it, while ^2^H-NMR reports directly on the bilayer itself. Compared to microscopy approaches, ^2^H-NMR offers high-resolution data not only specific to the lipid acyl chains themselves, but specific to acyl chain segments, i.e., those deep in the bilayer can be differentiated from those closer to the lipid headgroups.

Beyond deuterium NMR, it is also possible to use NMR to observe another isotope, ^31^P, to probe the effects of AMPs on phospholipid headgroups and other phosphorus-containing biomolecules in whole bacteria. Overall et al. [76] used static ^31^P cross-polarization (CP) NMR to observe *E. coli* treated with the AMP maculatin 1.1. They found it challenging to observe the ^31^P signal coming from the phospholipids, but quite interestingly, the AMP affected the dynamics of DNA inside the cell. This whole-cell NMR thus suggested a novel mechanism for maculatin 1.1 in disrupting an intracellular target. This and other findings from biophysical studies of whole cells [40] lead us to wonder how many other membrane-permeabilizing AMPs would be found to act by additional non-permeabilizing mechanisms if such mechanisms were more frequently investigated.

To directly probe the interactions between AMPs and lipids, REDOR NMR [77] gives distance measurements between NMR active nuclei, including distances from nuclei in AMPs to nuclei on lipids. REDOR distance measurements between ^15^N and ^13^C nuclei in the AMP maculatin 1.1 and ^31^P on the lipids in intact bacteria provided distances that were consistent with an α-helical peptide in a transmembrane orientation [78].

Complementary to the AMP-induced lipid-disruption information provided by ^2^H NMR of whole bacteria, flow cytometry [79], which can also be applied to intact bacteria, fills in a missing gap by indicating if the lipid-disruption observed by NMR is sufficient to permeabilize the bacteria to a fluorescent dye (Figure 1D). It also gives information on the granularity of the cell surface which may also be helpful in assessing AMP mechanism. Bacterial cells with a permeabilized cell membrane can be distinguished from intact bacterial cells using fluorescent dyes such as propidium iodide (PI) or SYTO 9. PI is a red-fluorescent dye that is non-permeable to the intact cell membrane and thus cannot enter viable cells but fluoresces upon entering the cells and binding DNA [79,80]. As seen in Figure 2B, the increase in lipid acyl chain disorder seen with 30% MSI-78 treatment is recapitulated in the flow cytometry where ~68% of the cells are pemeabilized to PI. Importantly, in these experiments the same cell and peptide concentrations were used in both NMR and flow cytometry, so that both types of experiments should have the same bound peptide-to-lipid ratio. While flow cytometry is very useful, it should be noted that it reports only indirectly on the state of the lipid bilayer by revealing if the bilayer has been perturbed enough to allow the dye to translocate across it, while ^2^H NMR provides direct read-out on the lipid acyl chains. Some of the current work in our group is aimed at comparing the effects of AMPs to CPPs on the lipid membrane disruption of whole cells, as judged by NMR and flow cytometry which give complementary—but distinct—information about the effects of AMPs on lipid membranes of whole cells. One of the findings (under review) from this work is that while flow cytometry confirms the CPP TP2 does not allow the dye to cross into the cell, it still has similar effects to AMPs on the ^2^H-NMR spectra, underlining a commonality in how AMPs and CPPs interact with lipid bilayers.

All-in-all, it is encouraging to see so many research groups employing whole-cell biophysical studies to better understand AMP mechanisms. This is bringing about a more complete understanding of this ubiquitous and fascinating class of molecules. Additionally, these efforts are informing work aimed at developing AMPs for clinical use, by pointing out that AMP optimization should be aimed at more interactions than just those between AMPs and lipids.

## Figures and Tables

**Figure 1 ijms-23-02740-f001:**
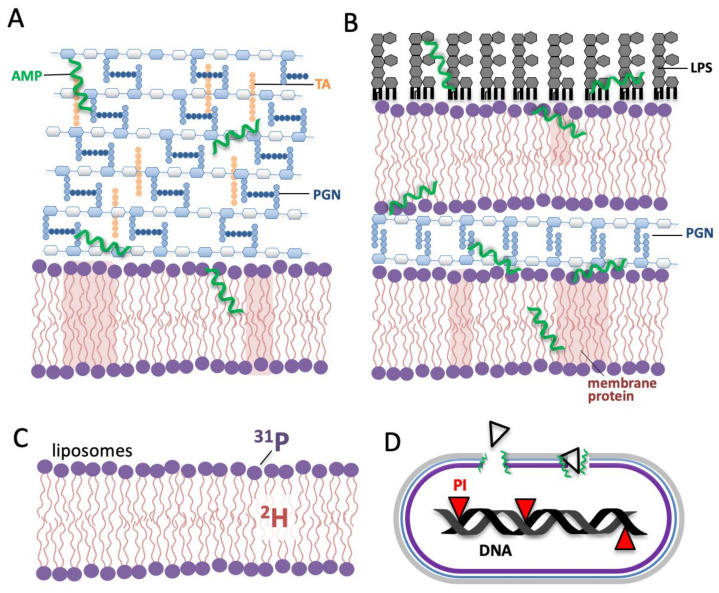
Selected techniques to study AMPs’ mechanisms in whole cells: NMR and flow-cytometry. (**A**) ^2^H NMR can be performed with membrane-deuterated Gram (+) and (**B**) Gram (−) bacteria. (**C**) NMR performed in liposomes uses ^2^H to indicate effects of AMPs on lipid acyl chains and ^31^P for the AMPs’ effect on lipid head groups. (**D**) Flow cytometry of PI-stained bacteria measures AMP-induced membrane permeabilization as PI only fluoresces if AMP permeabilizes the membrane enough for the PI to access the DNA.

**Figure 2 ijms-23-02740-f002:**
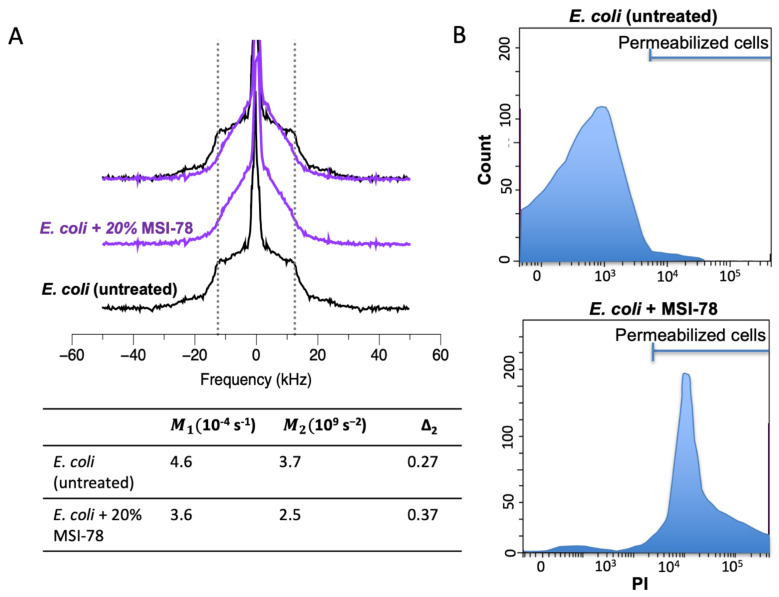
(**A**) ^2^H NMR spectra of membrane-deuterated *E. coli*. with (purple) and without (black) 20% AMP MSI-78 [68]. Dashed lines at ± 12.5 kHz are included to facilitate the comparison of the spectra. ^2^H NMR experiments were performed at 37 °C with a solid-state Bruker Avance II 600 MHz spectrometer, operating at a frequency of 92.15 MHz for ^2^H, with a triple resonance (HCD(N)) magic-angle spinning probe and 3.2 mm diameter rotor, without spinning. Moments and uncertainties (standard deviation in the mean value for three independently prepared samples) calculated from the spectra are shown below the spectra. (**B**) Schematic of flow cytometry cell count vs. PI fluorescence intensity for *E. coli* cells with and without MSI-78 can indicate if the disruption observed in the NMR spectra of the bacteria is sufficient to allow PI into the cells.

## Data Availability

The data presented in this study are openly available in Memorial University Dataverse: https://dataverse.scholarsportal.info/dataverse/memorial (accessed on 30 January 2022).

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
