# Peer review of "Antimicrobial Peptide Mechanisms Studied by Whole-Cell Deuterium NMR"

_ijms, 2022, doi:10.3390/ijms23052740_

Round 1
Reviewer 1 Report
Comments on paper: Antimicrobial Peptide Mechanisms Studied by Whole-Cell 2 NMR by Sarika Kumari et.al is a review intended to provide a comprehensive update on the use of whole-cell NMR technology on deciphering antimicrobial peptides and their mechanisms. This manuscript also deals with another technology flow cytometry and as a complementary method for measuring the permeabilization of AMP on bacterial membranes.
This study is interesting however it's not comprehensive as the manuscript dealt extensively with the AMP, their action, mechanism of interactions, but very little was provided in terms of in-cell NMR and very little information for use of flow cytometry techniques.
lines 206 to 220 needs to be expanded by providing the mechanisms of AMP cell penetration as designed and described by the authors and their references.
Authors should also explain the precautions taken to get better spectra and technological advances that came up recently supporting the interesting field of whole-cell NMR.
Also, authors should provide as to what advantages the technique is providing the scientists compared to other biophysical methods like fluorescence and microscopies.
Line 294: " Bacterial cells with a permeabilized cell membrane can be distinguished from intact bacterial cells using fluorescent dyes such as propidium iodide (PI) or SYTO 9." Authors have to explain can be studied using fluorescence microscopy and as to what advantages the whole-cell NMR provide.
Line 302: " Some of the current work in our group is aimed at comparing 302 the effects of AMPs to CPPs on the lipid membrane disruption of whole cells, as judged 303 by NMR and flow cytometry which gives complementary, but distinct information about 304 the effects of AMPs on lipid membranes of whole cells." it will be of utmost interests to the readers to see some casestudies from the authors to support the above quote.
Author Response
Reviewer: Comments on paper: Antimicrobial Peptide Mechanisms Studied by Whole-Cell 2 NMR by Sarika Kumari et.al is a review intended to provide a comprehensive update on the use of whole-cell NMR technology on deciphering antimicrobial peptides and their mechanisms. This manuscript also deals with another technology flow cytometry and as a complementary method for measuring the permeabilization of AMP on bacterial membranes.
This study is interesting however it's not comprehensive as the manuscript dealt extensively with the AMP, their action, mechanism of interactions, but very little was provided in terms of in-cell NMR and very little information for use of flow cytometry techniques.
Authors: We would like to thank the reviewer for the time and thought put into reviewing our manuscript. The work has been revised in response to the comments and we think this has led to a stronger paper.
First off, apologies, we’d been invited to write a review focusing on our own and closely related work with deuterium NMR of intact cells but the word “Deuterium” was inadvertently left out of the title. The title now reads, “Antimicrobial Peptide Mechanisms Studied by Whole-Cell Deuterium NMR”.
As the review article is currently at the suggested length, we have opted to point readers to tother important work in whole cell NMR by adding the following sentence and references (Lines 208 to 210 in the revised manuscript):
In addition to the studies highlighted below, the reader is directed to other works for a broader view of in-cell NMR including the exciting developments in solution NMR [64,65] and solid state 13C NMR [66,67].
[64] Luchinat, E., Banci, L. In-cell NMR: a topical review. IUCrJ 2017, 4, 108-118. https://doi.org/10.1107/S2052252516020625. 55
[65] Narasimhan, S., Folkers, G.E., Baldus, M. When Small becomes Too Big: Expanding the Use of In-Cell Solid-State NMR Spec troscopy. Chempluschem 2020, 85, 760-768. https://doi.org/10.1002/cplu.202000167. 56
[66] Nygaard, R., Romaniuk, J.A., Rice, D.M., Cegelski, L. Spectral snapshots of bacterial cell-wall composition and the influence of antibiotics by whole-cell NMR. Biophys J 2015, 108, 1380-1389 57
[67] Cegelski, L., O’Connor, R.D., Stueber, D., Singh, M., Poliks, B., Schaefer, J. Plant cell-wall cross-links by REDOR NMR spectroscopy. J Am Chem Soc 2010, 132, 16052-16057. 58
Reviewer: lines 206 to 220 needs to be expanded by providing the mechanisms of AMP cell penetration as designed and described by the authors and their references.
Authors: Please see lines 256 to 273 of the original paper (lines 290 to 312 of the revised paper) for this discussion. To help readers find this information and to better present the logical flow of the discussion, we have also added the following sentence (lines 384 to 444 of the revised manuscript):
“Before examining how whole cell 2H NMR speaks to the membrane perturbing mechanisms of AMPs, we first discuss the information contained in 2H NMR spectra of membanes in general.”
Reviewer: Authors should also explain the precautions taken to get better spectra and technological advances that came up recently supporting the interesting field of whole-cell NMR.
Authors: We have added a section to briefly summarize the key precautions and added a reference to point readers to the paper we have written specifically on the protocols for performing whole cell deuterium NMR (Lines 222 to 336 of the revised manuscript):
“In this and subsequent work, several factors have been identified that support the acquisition of reproducible 2H-NMR spectra from the bacteria, including adjusting the relative amounts of palmitic and oleic acids for the type of bacteria being grown, being very consistent with the growth and harvesting protocols used, transferring the cells into the NMR spectrometer quickly after growth, acquiring spectra in sequential blocks to monitor for changes in spectra over time, and using cell viability assays to assess how many bacteria are alive and metabolizing after their time in the NMR spectrometer [69,70].”
Reviewer 1: Also, authors should provide as to what advantages the technique is providing the scientists compared to other biophysical methods like fluorescence and microscopies.
Authors: Thanks for the prompt. In addition to the extra material based on the suggestions below, we have added the following paragraph (Lines 453 to 459):
2H NMR of intact cells provides complementary information to other techniques. For example, dye-leakage assays report indirectly on the bilayer behavior by indicating if the bilayer is perturbed enough to allow a dye to cross it, while 2H-NMR reports directly on the bilayer itself. Compared to microscopy approaches, 2H-NMR offers high resolution data not only specific to the lipid acyl chains themselves, but specific to acyl chain segments, i.e. segments deep in the bilayer can be differentiated from those closer to the lipid headgroups.”
Reviewer 1: Line 294: " Bacterial cells with a permeabilized cell membrane can be distinguished from intact bacterial cells using fluorescent dyes such as propidium iodide (PI) or SYTO 9." Authors have to explain can be studied using fluorescence microscopy and as to what advantages the whole-cell NMR provide.
Authors: We have added a sentence (Lines 488-491 in the revised manuscript) to clarify this:
“While flow cytometry is very useful, it should be noted that it only reports indirectly on the state of the lipid bilayer, by revealing if the bilayer has been perturbed enough to allow the dye to translocate across it, while 2H NMR provides direct read-out on the lipid acyl chains. “
Reviewer: Line 302: " Some of the current work in our group is aimed at comparing 302 the effects of AMPs to CPPs on the lipid membrane disruption of whole cells, as judged 303 by NMR and flow cytometry which gives complementary, but distinct information about 304 the effects of AMPs on lipid membranes of whole cells." it will be of utmost interests to the readers to see some case studies from the authors to support the above quote.
Authors: Thank you for your interest in this work. We are in the process of submitting a manuscript to BBA Biomembranes on this topic. We have added the following line to the review paper (Lines 508-512) to give a preview of the major findings. We feel it is not good practice to add more to the review since the work in question has not yet been peer reviewed.
“One of the findings (under review) from this work is that while the flow cytometry confirms the CPP TP2 does not allow the dye to cross into the cell, it still has similar effects to AMPs on the 2H-NMR spectra, underlining a commonality in how AMPs and CPPs interact with lipid bilayers.”
Reviewer 2 Report
This manuscript "Antimicrobial Peptide Mechanisms Studied by Whole-Cell NMR " is submitted as a review in IJMS
This is well written and the reading is pleasant.
Minot corrections should be done
L138: References are missing
L186, 188: Introductions of Figure 1C do not fit with the text, a proposal should be to add Figure 1C at the end of the previous sentence.
"By contrast, most investigations into the mechanism of AMP membrane perturbation, including NMR and fluorescence [20,56,57], employ only lipids rather than the entire bacterium (Figure 1C).
L206: The Davis group obtained the first 2H-NMR spectra of membrane-deuterated bacteria in the early 80s [58]. This is not true bacteria shoud be replace by liposomes.
Figure 2A the legends inside the table are not clear
Author Response
Reviewer: This manuscript "Antimicrobial Peptide Mechanisms Studied by Whole-Cell NMR " is submitted as a review in IJMS
This is well written, and the reading is pleasant.
Minot corrections should be done
Authors: We would like to thank the reviewer for the time and thought put into reviewing our manuscript. The work has been revised in response to the comments and we think this has led to a stronger paper.
Reviewer: L138: References are missing
Authors: The following references have been added, thank you
[50] Battista, F., Oliva, R., Del Vecchio, P., Winter, R., Petraccone, L. Insights into the Action Mechanism of the Antimicrobial Peptide Lasioglossin III. Int J Mol Sci 2021, 22, 2857.
[51] Oradd, G., Schmidtchen, A., Malmsten, M. Effects of peptide hydrophobicity on its incorporation in phospholipid membranes--an NMR and ellipsometry study. Biochim Biophys Acta 2011, 1808, 244-252.
Reviewer: L186, 188: Introductions of Figure 1C do not fit with the text, a proposal should be to add Figure 1C at the end of the previous sentence. By contrast, most investigations into the mechanism of AMP membrane perturbation, including NMR and fluorescence [20,56,57], employ only lipids rather than the entire bacterium (Figure 1C).
Authors: This sentence has been modified to:
“By contrast, most investigations into the mechanism of AMP membrane perturbation, including NMR and fluorescence [19,45,46], employ only lipids (Figure 1C) rather than the entire bacterium (Figure 1A,B).”
Reviewer: L206: The Davis group obtained the first 2H-NMR spectra of membrane-deuterated bacteria in the early 80s [58]. This is not true bacteria should be replaced by liposomes.
Authors: Apologies, we had the wrong reference here, it’s now been corrected to the one below. Note that this paper does actually have 2H-NMR of intact bacteria (Figure 4 in [68]).
[68] Davis, J.H., Nichol, C.P., Weeks, G., Bloom, M. Study of the cytoplasmic and outer membranes of Escherichia coli by deuterium magnetic resonance. Biochemistry 1979, 18, 2103-2112.
Reviewer: Figure 2A the legends inside the table are not clear
Authors: The table now has rows labelled “E. coli (untreated) and “E. coli (+30% MSI-78)” to be more clear
Reviewer 3 Report
The title of the review was “Antimicrobial Peptide Mechanisms Studied by Whole-Cell NMR”. In the manuscript, the author focused on the solid-state NMR. The liquid NMR was a necessary tool in-cell NMR research. The advances in research of liquid NMR should be added in this review.
The 2H and 31P spectra were discussed in the paper, and the spectrum of 31P should be provided. 13C solid state NMR was preferred method in the whole cell research. The introduction of the 13C should also be added in this paper.
The static solid-state NMR, magic angle spinning (MAS) NMR and cross-polarization (CP) NMR were introduced in the paper. As far as I know, the rotor kit should be used to analysis the whole cell. It would be better if the author could provide the information on instrument and probes.
Author Response
Authors: We would like to thank the reviewer for the time and thought put into reviewing our manuscript. The work has been revised in response to the comments and we think this has led to a stronger paper.
Reviewer: The title of the review was “Antimicrobial Peptide Mechanisms Studied by Whole-Cell NMR”. In the manuscript, the author focused on the solid-state NMR. The liquid NMR was a necessary tool in-cell NMR research. The advances in research of liquid NMR should be added in this review.
The 2H and 31P spectra were discussed in the paper, and the spectrum of 31P should be provided. 13C solid state NMR was preferred method in the whole cell research. The introduction of the 13C should also be added in this paper.
Authors: Apologies, we’d been invited to write a review focusing on our own and closely related work with deuterium NMR of intact cells but the word “Deuterium” was inadvertently left out of the title. The title now reads, “Antimicrobial Peptide Mechanisms Studied by Whole-Cell Deuterium NMR”. The work on whole cell solution NMR and 13C solid state NMR of whole cells is fascinating indeed. As the review article is currently at the suggested length, we have opted to point readers to this important work by adding the following sentence (Lines 208 to 210 in the revised manuscript) and references:
In addition to the studies highlighted below, the reader is directed to other works for a broader view of in-cell NMR including the exciting developments in solution NMR [66,67] and solid state 13C NMR [67,68].
[66] Luchinat, E., Banci, L. In-cell NMR: a topical review. IUCrJ 2017, 4, 108-118. https://doi.org/10.1107/S2052252516020625. 55
[67] Narasimhan, S., Folkers, G.E., Baldus, M. When Small becomes Too Big: Expanding the Use of In-Cell Solid-State NMR Spec troscopy. Chempluschem 2020, 85, 760-768. https://doi.org/10.1002/cplu.202000167. 56
[68] Nygaard, R., Romaniuk, J.A., Rice, D.M., Cegelski, L. Spectral snapshots of bacterial cell-wall composition and the influence of antibiotics by whole-cell NMR. Biophys J 2015, 108, 1380-1389 57
[69] Cegelski, L., O’Connor, R.D., Stueber, D., Singh, M., Poliks, B., Schaefer, J. Plant cell-wall cross-links by REDOR NMR spectroscopy. J Am Chem Soc 2010, 132, 16052-16057. 58
Since we have not performed 31P NMR on whole cells, we have cited other papers for this work (Lines 276-280 of the original paper, Lines 457-460 in the revised version):
“Overall et al. [77] used static 31P cross-polarization (CP) NMR to observe E. coli treated with the AMP maculatin 1.1. They found it challenging to observe the 31P signal coming from the phospholipids, but quite interestingly, the AMP affected the dynamics of DNA inside the cell. This whole-cell NMR thus suggested a novel mechanism for maculatin 1.1 in disrupting an intracellular target.”
Reviewer: The static solid-state NMR, magic angle spinning (MAS) NMR and cross-polarization (CP) NMR were introduced in the paper. As far as I know, the rotor kit should be used to analysis the whole cell. It would be better if the author could provide the information on instrument and probes.
Authors: The details have been added in figure 2 legend “2H NMR experiments were performed at 37°C with a solid-state Bruker Avance II 600 MHz spectrometer , operating at a frequency of 92.15 MHz for 2H, with a triple resonance (HCD(N)) magic-angle spinning probe and 3.2 mm diameter rotor without spinning.”
Round 2
Reviewer 1 Report
All comments are addressed by the authors
Reviewer 3 Report
Accept in present form.